# Unequal Protective Effects of Parental Educational Attainment on the Body Mass Index of Black and White Youth

**DOI:** 10.3390/ijerph16193641

**Published:** 2019-09-27

**Authors:** Shervin Assari, Shanika Boyce, Mohsen Bazargan, Ron Mincy, Cleopatra H. Caldwell

**Affiliations:** 1Department of Family Medicine, Charles R Drew University of Medicine and Science, Los Angeles, CA 90059, USA; Mohsenbazargan@cdrewu.edu; 2Department of Pediatrics, Charles R Drew University of Medicine and Science, Los Angeles, CA 90059, USA; ShanikaBoyce@cdrewu.edu; 3Department of Family Medicine, University of California Los Angeles, Los Angeles, CA 90095, USA; 4Center for Research on Fathers, Children, and Family Well-Being, Columbia University, New York, NY 10027-5927, USA; rmincysr@gmail.com; 5Columbia Population Research Center (CPRC), Columbia University, New York, NY 10027-5927, USA; 6Columbia University School of Social Work, Columbia University, New York, NY 10027-5927, USA; 7Center for Research on Ethnicity, Culture, and Health (CRECH), School of Public Health, University of Michigan, Ann Arbor, MI 48104, USA; cleoc@umich.edu; 8Department of Health Behavior and Health Education, School of Public Health, University of Michigan, Ann Arbor, MI 48104, USA

**Keywords:** population groups, race, whites, blacks, African-Americans, socioeconomic position, socioeconomic status, education, obesity, body mass index (BMI)

## Abstract

Background: Parental educational attainment is shown to be protective against health problems; the Minorities’ Diminished Returns theory, however, posits that these protective effects tend to be smaller for socially marginalized groups particularly blacks than whites. Aims: To explore racial differences in the effect of parental educational attainment on body mass index (BMI) in a national sample of US adolescents. Methods: In this cross-sectional study, we used baseline data of 10,701 (8678 white and 2023 black) 12–17 years old adolescents in the Population Assessment of Tobacco and Health (PATH; 2013). Parental educational attainment was the predictor. Youth BMI (based on self-reported weight and height) was the dependent variable. Age, gender, ethnicity, and family structure were covariates. Race was the focal moderator. Results: Overall, higher parental educational attainment was associated with lower youth BMI. Race, however, moderated the effect of parental educational attainment on BMI, suggesting that the protective effect of parental educational attainment on BMI is significantly smaller for black than white youth. Conclusions: In the United States, race alters the health gains that are expected to follow parental educational attainment. While white youth who are from highly educated families are fit, black youth have high BMI at all levels of parental educational attainment. This means, while the most socially privileged group, whites, gain the most health from their parental education, blacks, the least privileged group, gain the least. Economic, social, public, and health policymakers should be aware that health disparities are not all due to lower socioeconomic status (SES) of the disadvantaged group but also diminished returns of SES resources for them. Black–white health disparities exist across all high socioeconomic status (SES) levels.

## 1. Background

Body Mass Index (BMI) is now universally considered a marker of health [1,2]. High BMI is a predictor of premature mortality in the general population [3,4,5] and individuals with a chronic medical condition (CMC) [6].

Compared to white youth, black youth have higher BMI in the US [7,8,9]. Epidemiological studies have shown that education [9] and race [10,11] are independent predictors of BMI [9]. High BMI may mediate some of the inequalities and disparities in health [12,13,14,15,16,17,18,19,20,21,22,23]. Finally, race, ethnicity, and economic factors have some separate, additive, or multiplicative effects [9]. Educational attainment is one of the main predictors of BMI [9].

Similarly, black youth have a higher risk of overweight and obesity compared to their white counterparts [24,25,26]. Disparities in BMI are among the main contributors to racial health disparities in other health domains including but not limited to chronic disease and mortality [27,28]. Inequality in BMI in youth is also a gateway to the very persistent disparities in the burden of obesity seen in adults [7]. Thus, there is a need to understand the socioeconomic factors that explain racial differences in BMI [29], particularly those between black and white youth [30].

Given the protective effect of high socioeconomic status (SES) against high BMI in youth [31,32,33], adults [34,35], and older adults [36,37], SES is one of the best candidates as an explanatory factor for explaining the BMI differences between blacks and whites [30]. This is particularly relevant given the literature that has linked low SES to higher BMI among black youth [32,38,39,40]. Among various SES indicators, parental educational attainment seems to be one of the main protective factors against high BMI and obesity [41,42,43].

Minorities’ Diminished Returns (MDRs) [44,45] proposes that at least some of the racial/ethnic disparities in BMI are due to less than expected protective effects of SES on BMI for the black community [32,39,46]. This suggests: (a) Racial disparities in obesity and high BMI are not all due to SES gaps across racial groups but also because of the differential health gains that follow SES resources such as parental education for black populations, and (b) the relative racial gap in obesity and BMI widens as SES increases [32,39,46], which emphasizes a need to study and address racial disparities in obesity across all SES levels, not just among low SES groups [44,45].

In a cross-sectional study that used data from the National Survey of Children’s Health (NSCH, 2003–2004), 67,610 white and 9095 black children of 2–17 years old were compared for the effect of parental income on childhood obesity. This study showed that family income reduces the odds of obesity for white but not black children [32]. In a 1-year follow up study of urban families, race by gender groups were compared to the effects of parental educational attainment at birth on future BMI at age 15. In this study, the most consistent protective effects of parental education were found for the effects of family SES against high future BMI for white girls, followed by white boys. No association was found for maternal education, family structure, and family income at birth and future youth BMI at age 15 for black boys or girls. However, this study did not use a nationally representative sample, thus the results could not be generalized to the US population [46]. Thus, despite the existing evidence that suggests MDRs may apply to racial differences in BMI of youth, there is still a need to test this hypothesis in nationally representative data sets that generate representative results.

We conducted this study to test whether race shows an interaction with parental educational attainment on youth BMI in the US. In line with the MDRs literature [32,39,46], we hypothesized that the protective effect of parental educational attainment would be significantly smaller for black than white youth. Conceptualizing race as a social rather than a biological factor, we argue that MDRs are not because of groups’ or individuals’ inherent differences in their ability to translate their resources but the differential treatment of the society that is associated with marginalization and stigmatization of blacks [44,45]. As a result of unequal life circumstances, we expect to observe MDRs for the effects of parental educational attainment on BMI in blacks than whites.

## 2. Methods

### 2.1. Design and Settings

This is a secondary analysis of wave 1 of the Population Assessment of Tobacco and Health (PATH) youth data. Funded by the NIH and FDA, PATH is a state-of-the-art study on health problems such as tobacco use, substance use, and related behavioral issues among US adolescents and adults. Overall, PATH has enrolled 53,178 individuals who were 12 years or older at baseline. From this sample, 13,650 were youth (12–17 years old). Wave 1 data were collected between 2013 and 2014. Although PATH has also recruited adults, this analysis is only focused on youth. We were interested in the association between SES and BMI rather than a change in BMI, we limited our analysis to cross-sectional data (Wave 1) of the PATH study. We used the publicly available PATH data set.

### 2.2. Sample and Sampling

The PATH study’s population of interest in Wave 1 was the civilian, non-institutionalized US population of 12–17 years old individuals in the US. The PATH study used a four-stage stratified area probability sample design. Stage one was the selection of a stratified sample of geographical primary sampling units (156 PSUs). These PSUs were either a county or a group of counties. Stage two formed and sampled smaller geographical segments in each PSU. Stage three sampled residential addresses, using the US Postal Service Computerized Delivery Sequence Files. The fourth stage was the selection of one person from each sampled household.

### 2.3. Analytical Sample

The current analysis is limited to youth who had complete data on age, gender, parental education, race, ethnicity, and BMI. Our final analytical sample was 10,701 (8678 white and 2023 black) adolescents. For the purpose of this study, the sample was specifically restricted to white and black respondents. (i.e., Asians and people of other races were excluded).

### 2.4. Study Variables

The study variables include age, gender, race, ethnicity, parental educational attainment, marital status of the parents (family structure), and BMI.

#### 2.4.1. Demographic Factors 

Age was a dichotomous variable as below: (1) 12 to 15 years old, and (2) 16 to 17 years old. Public PATH data set does not provide more granular data on age.

#### 2.4.2. Race and Ethnicity 

Race was self-identified and operationalized as a dichotomous variable: blacks versus whites. Ethnicity was also self-identified and operationalized as a dichotomous variable: Non-Hispanic versus Hispanic.

#### 2.4.3. Socioeconomic Status 

Parental educational attainment was a five-level variable as below: (1) less than high school, (2) high school graduate or equivalent, (3) some college including no degree or an associate degree, (4) Bachelor’s degree, and (5) advanced degree. This was a continuous measure ranging between 1 and 5.

#### 2.4.4. Body Mass Index (BMI) 

The study calculated the BMI using participants’ height and weight. First, height and weight were recorded in feet/inches and pounds, respectively. In the second step, height and weight were calculated in meters and kilograms. BMI was calculated by dividing weight (kilograms) by height squared (meters squared).

### 2.5. Conceptual Model

Built on the MDRs, our study is mainly focused on the interaction between race and parental educational attainment (SES). As a result, the main predictor of interest is parental educational attainment. The outcome of interest is BMI. The moderating variable is race. We expect high parental educational attainment to be associated with lower BMI in the overall sample, however, we expect this effect to be weaker in black compared to white youth. We call weaker effects of SES indicators in the non-white group as diminished returns of SES.

### 2.6. Data Analytical Plan

We analyzed the data using SPSS 23.0 (IBM Corporation, Armonk, NY, USA). We applied survey design weights. SPSS uses a Taylor series linearization to re-estimate standard errors (SE) of the survey data. For bivariate analysis, we tested the bivariate correlations between our study variable using the Pearson correlation test. No two variables had correlation coefficients equal or larger than 0.50 so we ruled out any collinearity between our independent variables. This strategy was taken for the full sample and also racial groups.

For multivariable analysis, we applied linear regression models. Overall, we ran four models. We ran the first two models in the pooled sample. Then we ran two additional models across racial groups. *Model 1* did not have the interaction term. *Model 2* included the interaction term. Model 3 was specific to white youth. Model 4 was specific to black youth. From our models, we reported regression coefficient (b), Standard Error (SE), 95% Confidence Interval (CI), and *p*-value. We did not impute data. Complete data cases were entered into our analysis.

We tested the assumptions and requirements for linear regression models. This includes ruling out the multicollinearity between the independent variables. We also confirmed the near to normal distribution of error terms of our linear regression models.

We did not control for income for methodological and conceptual reasons. Some evidence suggests that income may be the reason educational attainment does not generate the very same health across racial and ethnic groups [47,48]. Thus, controlling income would be adjusting for the mediator, which introduces bias due to controlling for the intermediate variable [49]. Other evidence suggests that MDRs are more relevant to educational attainment than income because less societal processes interfere with families’ abilities to use their income, compared societal barriers [47]. Thus, some evidence suggests that MDRs are most relevant to the distal (e.g., educational attainment) than proximal (e.g., income) SES indicators [47,50].

### 2.7. Ethics

All youth who participated in the PATH study provided assent. All their parents/guardians/caregivers provided written informed consent. The PATH study protocol was approved by the Westat institutional review board.

## 3. Results

### 3.1. Descriptive Statistics

This study included 10,701 US adolescents who were either white (*n* = 8,678, 81.1%) or black (*n* = 2023, 19.9%).

Table 1 shows descriptive statistics of the complete sample by race. White youth were more likely to be of Hispanic ethnicity (25% vs. 9.9%, *p* <0.01), have parents with higher educational attainment (2.89 vs. 2.62, *p* <0.01), and married parents (69.6% vs. 38.9%, *p* <0.01). We did not find significant differences in age and gender between white and black youth. On average, black youth had higher BMI than white youth (23.30 vs. 22.36, *p* < 0.01) (Table 1).

### 3.2. Multivariable Models in the Pooled Sample

Table 2 presents the summary of the results of two linear regression models with parental educational attainment as the independent variable and BMI as the dependent variable. Both models were estimated in the complete sample. *Model 1* was the non-interactive model that entered the main effects of educational attainment and race in addition to the covariates. *Model 2* also added an interaction term between race and educational attainment. Covariates were the same in these two models.

Based on *Model 1*, high educational attainment was associated with lower BMI. In this model, being black was also associated with higher BMI. *Model 2* revealed an interaction between race and parental educational attainment on BMI, suggesting that high parental educational attainment has a smaller protective effect on BMI for black than white youth. This was evident by a regression coefficient which was negative for the main effect of parental educational attainment (*Model 1*) and a regression coefficient which was positive for the interaction term between race and parental educational attainment (*Model 2*) (Table 2).

### 3.3. Multivariable Models by Race

Table 3 presents the results of two linear regression models with educational attainment as the independent variable and BMI as the dependent variable. These models were set up similarly, and each one was run in one racial group. *Model 3* and *Model 4* were estimated in whites and blacks, respectively.

Based on *Model 3*, high educational attainment was associated with lower BMI in whites. *Model 4*, however, did not show the same pattern of association for blacks. That is, high educational attainment showed a statistically significant protective effect on BMI for whites but not blacks (Table 3).

Figure 1 shows the bivariate correlation between parental educational attainment (X-axis) and BMI (Y-axis) in white and black youth. As this figure shows, there was a negative (inverse) correlation between parental educational attainment and BMI in white but not black youth.

## 4. Discussion

The current study showed two main findings. First, overall, higher parental educational attainment was associated with lower BMI in US youth. Second, race showed a significant interaction with parental educational attainment suggesting that high parental educational attainment has a smaller protective effect against high BMI for black than white youth. In fact, the association was only significant for whites but not blacks.

Our first finding, the inverse association between family SES and youth BMI, is in line with the extensive literature on fundamental causes [51,52,53] and social determinants of health (SDOH) [54,55,56]. A large body of research has shown that parental educational attainment is associated with better health and well-being [41,57,58,59] and lower risk of obesity [60].

Our previous work on MDRs shows black youth from high SES remain at an increased risk of high BMI compared to high SES whites [32,39,46], a pattern which is seen for a wide range of SES indicators, health outcomes, youth, adults, and older adults [61,62,63,64]. Similar patterns are reported for educational attainment [39,46], income [32,65], employment [66], and marital status [67] on obesity [39,46], depression [65], anxiety [67], self-rated health [68,69], chronic disease [70], and even mortality [66]. In several papers, smaller effects are shown for blacks than whites [32,39,46].

MDRs may be due to social stratification that operates through biological mechanisms such as those involved in responding to stress (e.g., allostatic load, weathering, metabolic syndrome) [71]. There is, however, a need for further studies to understand the role of biological and physiological pathways that may explain MDRs on BMI [72,73,74,75,76,77,78,79].

We need to better understand the role of unsafe environment [80], food deserts [81,82,83,84], density of fast food restaurants [85,86,87,88], and availability of healthy food options [89] across locations based on the intersection of race and SES. Educational attainment has smaller effects to increase blacks’ employment and enhance their living conditions, compared to whites [57,58,90,91]. For example, a larger proportion of blacks with high educational attainment remain under poverty [90]. We argue that residential segregation, racism, labor market discrimination, and limited resources in predominantly black schools all contribute to MDRs of educational attainment on BMI [46].

We see race as a political and social rather than a biological construct [92]. Social stratification by race is shaped by history, politics, and laws [93,94]. Through political power, race is a strong proxy of access to the societal resources and opportunities [92]. For example, marginalization of black men, through mass incarceration and the war on drugs, has had a strong political impact in who gets elected, and who writes the laws [92]. Political power is a major determinant of health disparities [92,93,95,96]. Social and public policies, traditionally written by policymakers and politicians, have maintained structural inequalities in the United States. Reducing racial stratification needs considerable political will. Policymakers should target the very structural context of the U.S. if we want to eliminate health inequalities [92,93,95,96]. Today’s highly polarized political environment, however, is a major barrier against reaching a bipartisan agreement on policies that are needed to solve the structural health inequalities. Unfortunately, good-faith policy intentions have failed to effectively address health disparities. This is in part because of the differential effect of resources on health without addressing structural forces [92,93,95,96].

### 4.1. Implications

The results may have some implications for public, economic, social, and health policies. First, we argue that policymakers should not assume that SES indicators are equally protective against health problems including high BMI for various racial groups. As new policies are being implemented, there is a need for “impact analysis” to track effects on the population overall, as well as sub-population differences in such effects. Given the limited benefit of high SES on BMI among black youths, providing incentives for health in predominantly black areas may be required. An enhancement to the quality of education and schooling in predominantly black areas are also needed. We need programs that enable black communities, with the translation of educational attainment, to obtain tangible health outcomes. Finally, the elimination of racism and segregation is needed if we want to help black families be able to turn their educational attainment to health outcomes. Societal interventions, through public policies, would probably be more effective than individual-level interventions that overemphasize individual choices and preferences and ignore social stratification [97].

National and local policies are needed to reduce the racial/ethnic and SES disparities in BMI, including those that are due to MDRs [45,46,50,62,63,64,67,68,98,99]. Additional investment is needed in reducing the societal risk factors of obesity in urban black communities. There is a need to increase the availability of healthy food, walkable spaces, green spaces and parks, and other societal resources that can help black families and youth to avoid obesity. To undo racial and ethnic disparities in obesity and related morbidity, structural changes are needed that equalize environmental opportunities for maintaining healthy BMI.

### 4.2. Future Research

There is also a need to conduct further studies that help us understand, based on race, why some youth with highly educated parents remain susceptible to obesity and high BMI. Research should assess how the density of healthy and unhealthy food, the walkability of living areas, availability of green spaces and parks, and other structural determinants of obesity, mediate or moderate the MDRs observed in this study. There is also a need to test the efficacy of various policies in reducing MDR-related disparities in health.

### 4.3. Limitations

This study had some methodological and conceptual limitations. First, given the cross-sectional design of our data, causal inferences are not plausible. The sample size was imbalanced across racial groups, which is a common pattern in national surveys. In addition, we did not have access to a wide range of other SES indicators of the household such as income, employment, and occupation type. Finally, this study used an over-simplistic measure of BMI. Several other indicators such as waist to hip ratio, sub-cutaneous adiposity, and other indicators need to be studied. The results should be interpreted with knowing that we did not compare non-Hispanic white and non-Hispanic blacks but all groups of whites and blacks. However, our robustness analysis did not show any difference in the interaction term when we limited the sample to non-Hispanic whites and blacks. Finally, we did not include area-level SES indicators. Despite these limitations, we believe that this study extends the existing literature on race, SES, and BMI. We had previously shown similar findings in the Fragile Families and Child Well-being Study (FFCWS) data. However, FFCWS was a non-representative study composed of economically at-risk families who were only selected from large cities. The results of the current study, however, are generalizable to the US youth sample, given the nationally representative nature of the sample.

## 5. Conclusions

In the United States, racial minority status limits the health gains that follow parental educational attainment. While high parental educational attainment helps people stay healthy and avoid high risk behaviors such as smoking, the most privileged group gains the most and the least privileged groups gain the least from such potential. As a result, we should expect an additional risk of high BMI and obesity in middle-class blacks. Researchers should not take a minimalistic and over-simplistic approach by focusing on health disparities among blacks of low SES. They should consider that health inequalities and disparities can be observed across all SES levels, and the SES gap is only a part of the story. Given that health disparities are also faced by middle-class blacks, health disparity solutions should focus on other sections of black communities, that may not be low SES but still have increased risk of poor health outcomes. Policymakers should also go beyond equalizing SES and address the barriers that are more common in the lives of blacks and hinder their ability to turn their available resources into tangible health outcomes.

## Figures and Tables

**Figure 1 ijerph-16-03641-f001:**
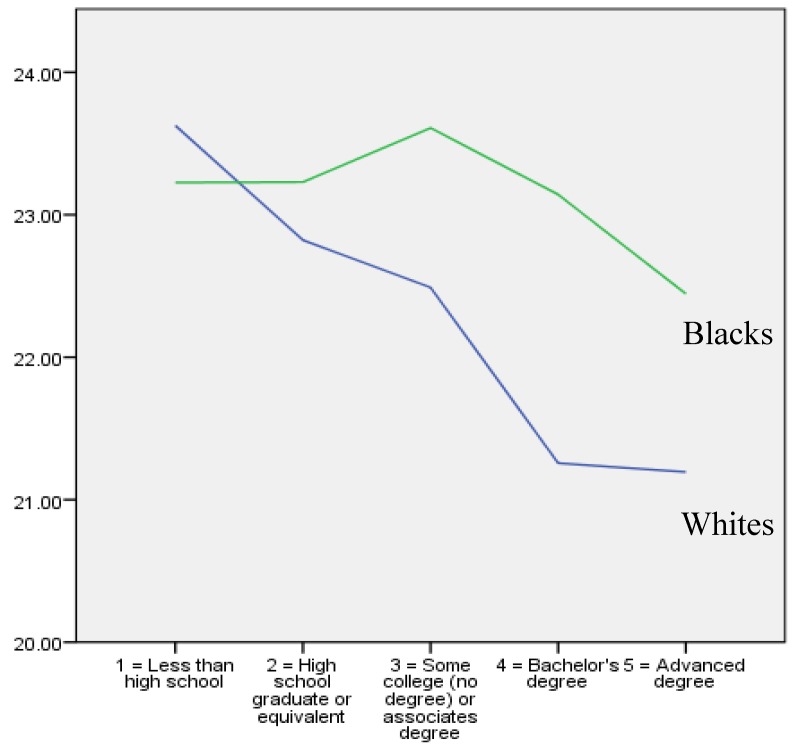
Average youth body mass index (BMI) based on parental educational attainment in whites (**Blue line**) and blacks (**Green line**).

**Table 1 ijerph-16-03641-t001:** Descriptive statistics in the complete sample and by race.

Characteristics	All		Whites		Blacks	
	n	%	n	%	n	%
Ethnicity *^,a^						
Non-Hispanic	8179	77.8	6407	75.0	1772	90.1
Hispanic	2329	22.2	2135	25.0	194	9.9
Age						
12–15	5474	51.2	4437	51.1	1037	51.3
16–18	5227	48.8	4241	48.9	986	48.7
Gender						
Women	5143	48.2	4180	48.3	963	47.9
Men	5531	51.8	4483	51.7	1048	52.1
Marital Status *^,a^						
Not Married	3870	36.2	2635	30.4	1235	61.1
Married	6817	63.8	6030	69.6	787	38.9
	**Mean**	**SD**	**Mean**	**SD**	**Mean**	**SD**
Parental educational attainment (1–5) *^,b^	2.84	1.22	2.89	1.23	2.62	1.16
Body Mass Index (BMI) *^,b^	22.53	5.25	22.36	5.06	23.30	5.92

* *p* < 0.01, ^a^ Chi square test for comparison of black and white youth. ^b^ Independent samples *t*-test for comparison of means between black and white youth.

**Table 2 ijerph-16-03641-t002:** Summary of linear regressions on body mass index (BMI) in the pooled sample.

	b	SE	95% CI	*p*
**Model 1 (All, Non-interactive Model)**				
Race (Blacks)	0.84	0.13	0.58–1.10	<0.001
Parental Educational Attainment (1–5)	−0.45	0.04	−0.54–−0.37	<0.001
Ethnicity (Hispanics)	0.64	0.13	0.39–0.89	<0.001
Parents Married	−0.36	0.11	−0.57–−0.15	0.001
Gender (Boys)	0.00	0.10	−0.19–0.20	0.987
Age (16–18 Years Old)	1.79	0.10	1.60–1.99	<0.001
Constant	22.88	0.17	22.54–23.23	<0.001
**Model 1 (All, Interactive Model)**				
Race (Blacks)	−0.65	0.33	−1.29–−0.01	0.046
Parental Educational Attainment (1–5)	−0.55	0.05	−0.65–−0.46	<0.001
Race (Black) × Parental educational attainment	0.55	0.11	0.34–0.77	<0.001
Ethnicity (Hispanics)	0.57	0.13	0.32–0.82	<0.001
Parents Married	−0.36	0.11	−0.57–−0.14	0.001
Gender (Boys)	0.00	0.10	−0.19–0.20	0.979
Age (16–18 Years Old)	1.78	0.10	1.59–1.98	<0.001
Constant	23.20	0.18	22.83–23.56	<0.001

B: Regression Coefficient CI: Confidence Interval; SE: Standard Error.

**Table 3 ijerph-16-03641-t003:** Summary of linear regression models on body mass index (BMI) by race.

	b	SE	95% CI	*p*
**Model 3 (Whites)**				
Parental Educational Attainment (1–5)	−0.54	0.05	−0.64–−0.45	<0.001
Ethnicity (Hispanics)	0.62	0.13	0.37–0.88	<0.001
Parents Married	−0.41	0.12	−0.64–−0.18	0.001
Gender (Boys)	0.16	0.11	−0.05–0.37	0.139
Age (16-18 Years Old)	1.82	0.11	1.61–2.03	<0.001
Constant	23.09	0.19	22.73–23.46	<0.001
**Model 4 (Blacks)**				
Parental Educational Attainment (1–5)	−0.02	0.11	−0.24–0.21	0.869
Ethnicity (Hispanics)	0.11	0.45	−0.76–0.99	0.804
Parents Married	−0.17	0.27	−0.71–0.36	0.529
Gender (Boys)	−0.67	0.26	−1.18–−0.15	0.012
Age (16-18 Years Old)	1.64	0.26	1.12–2.16	<0.001
Constant	22.98	0.39	22.22–23.74	<0.001

B: Regression Coefficient CI: Confidence Interval; SE: Standard Error.

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
