# Peer review of "Unequal Protective Effects of Parental Educational Attainment on the Body Mass Index of Black and White Youth"

_ijerph, 2019, doi:10.3390/ijerph16193641_

Round 1
Reviewer 1 Report
The paper uses the PATH survey series to test the hypothesis that the effects on BMI originating from socioeconomic resources during childhood/adolescence like parental education are moderated by the race of the individual. They use the framework of MDR’s theory to produce an econometric test of the hypothesis, which is, in my opinion, very much in line with most of the hypothesis testing developed in established MDR research. Results of the paper are in line, as well, with much of what has been done in this area of study. The authors, however, discuss their findings in light of possible policy solutions (stepping a little bit aside from a medical approach to MDR while emphasizing social aspects related to racial disparities in health).
Overall, this is a paper that needs a heavy editing. Their results and their interpretation as well as their models need to be better outlined and described. That said, I think the paper has an interesting finding, and it also expands its findings to the arena of policy, making the paper suitable to a wider range of readers from different discipline traditions. I also think that it fits well with the research published in the IJERPH. I would give it a R&R with major revisions. My observations follow:
Data, methods, and results:
· The section “Design and settings” includes a lot of non-essential information about the data (e.g., that the authors downloaded it from the ICPSR, list of funders, and so forth). Remove it.
· In the section “Sample and sampling” the authors mention the PATH study used a four-stage stratified area probability sample design. Did they account for sampling design? If they can, they should use the 156 PSUs to account for the survey design (this can be easily done in many statistical software). This should be done at least as a robustness check since survey designs can easily affect biasedness and precision of estimates.
· In the section “Analytic sample” the authors mention “The current analysis is limited to youth who had valid data.” What do they mean by “valid data? Do they refer to complete data cases? If so, what about missing data? How did they handle missing data? These issues need to be spelled out.
· Interestingly, the authors did not control for parental income. Why was that? Do they think that parental income is a mediator of parental education? If so, explain.
· It is a little bit confusing, but it seems the authors split the sample in two age groups (12-15, and 16-17 yrs). If so, what is particular about these two groups that make them suitable for partitioning the sample? If this is not the case, why did they code the variable age in this manner? Why not just use it as a continuous variable? Explain.
· Under “race and ethnicity” it is not clear (as stated) if they will use 2 or 4 groups (this only becomes evident once one arrives at the Results section). It is not clear if there are Hispanic and non-Hispanic blacks as well as Hispanic and non-Hispanic whites.
o It is different to run an analysis on non-Hispanic blacks and non-Hispanic whites than running the analysis on all blacks and all whites controlling for Hispanic origin (which is the approach here). At least as a robustness check, they should run their analysis using non-Hispanic blacks and non-Hispanic whites.
· One big issue with using BMI as a continuous dependent variable, is that BMI has different clinical implications at different points of the scale. It is unhealthy when BMI is at both the lower and upper tails of the BMI distribution. In other words, to increase BMI at the lower tail is actually desirable; to increase it at the upper tail is not. But this is not captured in the current analysis. Yet, it seems the authors are concerned about BMI in the context of obesity, not on BMI as a continuous, monotonic scale (which is an incorrect operationalization of the scale).
o The authors need to either re-focus their study or judiciously elaborate a justification about their use of the dependent variable as is.
· The authors state their main predictor of interest is SES (line 109). Yet, if the authors have an interactive term, they cannot differentiate the effect of parental education from that of race. This means that their main predictor is not SES alone, but the interaction itself.
o Also, they do not have a variable for “SES”. Their variable is parental education, not SES.
· Instead of writing “Data analytical plan”, write “methods”. Some observations follow:
o Get rid of “(IBM Corporation, Armonk, NY, USA)”.
o Get rid of “First, we examined the distribution of our categorical and continuous variables to rule out multi-collinearity between independent variables. For univariate analysis, we used frequency tables as well as means (SD). For bivariate analysis, we used Pearson correlation test.” This information is part of the researcher’s data analysis preceding the statistical analysis and provides non-essential information about the statistical analysis.
o Get rid of “From our models, we reported b, SE.” This is a poorly elaborated phrase. Maybe write something like: “Table X lists the parameter estimates from our regression models”?
· Does the “ethics” section need to be in the body of the text? It usually goes at the end of the manuscript, and it conforms to stipulations of the publishing journals, not to stipulations of the authors (unless they themselves generated and collected the data).
· Under the section “Results”:
o The first phrase is already stated in the paper. Cut it, and start the paragraph with “Table 1 shows…”
o The following phrase is incorrect: “White and Black youth did not vary in age and gender.” Of course they vary; age and gender are variables. Maybe what the authors want to say is that there are no statistically significant differences in the mean age and gender composition between whites and blacks in the sample?
o When making statements like “Black youth had higher BMI than White youth”, write “on average” at the end—which is what I think the authors want to state.
o In Table 1, the columns for white and black under the rows white and black, are redundant (that same information is the one listed in the first column).
o In Table 1, at the bottom, there is a footnote about statistical significance, a chi-squared test, and an independent samples test. Remove that information. Unless the authors plan to report some statistical testing (which they don’t in the table), they should get rid of the footnote.
o The “bivariate analysis” as well as Table 2 are unnecessary. They provide no insight into the research question or the hypotheses being tested. They also do not add any insight into the statistical analysis. They should be removed.
o Table 3 is not clear. It is stated “Model 1 only entered the main effects of educational attainment and race” and yet I see many other coefficients listed in Table 3 under Model 1. What I think the authors are trying to say is that Model 1 is a non-interactive model whereas Model 2 adds the race-by-education interaction term to Model 1. Models 1 and 2 share the same set of covariates, correct?
§ Another confusion about Table 3 is that it has a column for “B” which I imagine this is the coefficient (?), and also a column for “OR” for an odds ratio… of what? This is not clear.
· Importantly, statistical tests for coefficients and odds ratios are different. The baseline for the null hypothesis in a coefficient is “0” whereas for an odds ratio is “1”. To put them together in the same table, one at the side of the other is confusing (unless in the context of methods for binary data). Readers without much statistical training will not understand if the SEs, 95%CI, and p-values are for the coefficients or for the odds ratios. Remove the odds ratios. This applies for tables 3 and 4.
§ Reorganize Table 3: First list the coefficients of interest, then the confounders.
o The interpretation of the interaction in Model 2 is confusing (lines 151-155). In particular this phrase: “This was evident by a negative b for the main effect of parental educational attainment and then a positive b for the interaction term between race and parental educational attainment.”
§ First, please don’t say “negative b” or “positive b”. Just say a coefficient is negative or positive.
§ The phrase says “[…] and then a positive…”. It seems as if, first a model was fitted and then another (?). Or is it that the authors are going from Model 1 to Model 2 (?). This is confusing.
§ Interpretation of coefficients in interactive models depends on the complete model specification. This is not done here and can potentially lead to misinterpretations of the coefficients.
o The multivariable models are “by” not “in” race/ethnic groups (line 160).
§ Mistake: I see that the subtitle says “in ethnic groups” (line 160) but the regressions seem to be by race, not by ethnicity.
§ I don’t see what additional information these models bring aside from that already in Table 3. The difference is that, models in Table 4, are “full-interactive” models whereas those in Table 3 are not. Models in Table 4 may be used as robustness check; I don’t see a valid justification to include them in the paper. If the authors want to describe the separate effects, and to test for their respective statistical significance, for whites and blacks, they can perfectly do that using Model 2 in Table 3. Plus—again—models in tables 3 and 4 are not comparable because models in Table 4 are fully-interactive. To be explicit: By separating the models in Table 4, they are intrinsically specified as if all variables in the right side of the equation are interacted by the respective race they were ran for (whereas in Table 3 only race and parental education are interacted). The models, therefore, follow different econometric specifications; implications from Table 4 do not map into those from models in Table 3.
Theoretical context:
· The authors need to make some effort to spell out what is the unique contribution of this piece, considering that much similar research has been done in MDR. If not focused appropriately, this paper can be just one more of many—a replication exercise. Yet, I think the paper contains valuable information that may go beyond a simple replication of previous research.
· The Background section is too short. I was felt with the impression it went midway. If space is no constraint, I would advise to expand it.
· The Implications section is a little repetitive. Edit it.
· Importantly, the authors want to stress that their results bring implications for a wide range of policy arenas, from public to economic and from social to health-specific policy. I really appreciate this context, especially because research has shown again and again that in spite of good-faith policy intentions, we cannot effectively address health disparities and the differential effect of resources on health without talking about structural forces. Yet, I still believe the authors’ approach is still underdeveloped. Social and health policy is written by politicians, and today’s political environment is highly polarized, there is gridlock in Congress, state legislatures behave along party lines, presidents and presidential candidates capitalize on totally different constituencies, lobbies, and corporate interests, and so forth. The paper would greatly benefit from outlining in more detail a structural context to their findings.
o Why we observe the health differences we observe by race and, specifically, the BMI differences the authors report by parental education and race? Yes, we know the reasons go along racism, discrimination, differential treatment, and so forth. But these are the proximal causes. Again, I think the paper would greatly benefit if the authors address fundamental causes, and engrain their findings within the theoretical framework of the historical, legal, and political determinants of the social determinants of health (like parental education) and how they are differently distributed in the population by race.
o There is also research they can incorporate on social stratification and biological mechanisms that respond to stress (e.g., allostatic load, weathering, metabolic syndrome) that fit very well with their findings. And, again, I think the paper would benefit from signaling that social stratification by race is historically, politically, and legally patterned.
· The authors mention skin color, but skin color (as a variable) did not enter in the equation. Their variable is “race” and, unless they want to further elaborate on the idea of skin color (as a phenotypic expression) above and beyond race as a lived experience, I would recommend avoiding mentioning it in the paper. And, again, there is research pointing to race as a political construction (as well as the distribution of the social determinants of health is political, too).
Other minor observations:
· Lines 114-115: Write “We applied PATH’s survey design weights”.
· Why the authors used the Taylor series linearization to re-estimate the standard errors? Elaborate and justify.
· Line 59: spacing: MDRs literature[8,15,22] ,
· Line 70: spacing: state -of-the art
· Line 89: typo: 1,0701 instead of 10,701.
· Line 93: the parenthesis in “(age and gender”) should also include “race and ethnicity”
· Line 95: typo or confusing word choice: “as below”
· Line 148: Write: “were estimated using the complete sample” (not “in the overall sample”). The same goes for Table 3’s title: “using the complete sample”.
Author Response
The paper uses the PATH survey series to test the hypothesis that the effects on BMI originating from socioeconomic resources during childhood/adolescence like parental education are moderated by the race of the individual. They use the framework of MDR’s theory to produce an econometric test of the hypothesis, which is, in my opinion, very much in line with most of the hypothesis testing developed in established MDR research. Results of the paper are in line, as well, with much of what has been done in this area of study. The authors, however, discuss their findings in light of possible policy solutions (stepping a little bit aside from a medical approach to MDR while emphasizing social aspects related to racial disparities in health).
Thank you!
Overall, this is a paper that needs a heavy editing. Their results and their interpretation as well as their models need to be better outlined and described. That said, I think the paper has an interesting finding, and it also expands its findings to the arena of policy, making the paper suitable to a wider range of readers from different discipline traditions. I also think that it fits well with the research published in the IJERPH. I would give it a R&R with major revisions. My observations follow:
Thank you! As the changes are shown in yellow, we heavily revised our paper. Thanks for the R&R.
Data, methods, and results:
· In the section “Sample and sampling” the authors mention the PATH study used a four-stage stratified area probability sample design. Did they account for sampling design? If they can, they should use the 156 PSUs to account for the survey design (this can be easily done in many statistical software). This should be done at least as a robustness check since survey designs can easily affect biasedness and precision of estimates.
Thank you! Yes, of course we have adjusted the survey weights. This is explained in the statistical note. This is also why Taylor series linearization is used.
· Interestingly, the authors did not control for parental income. Why was that? Do they think that parental income is a mediator of parental education? If so, explain.
Thank you! Yes, there is a major conceptual reason for why we have decided not to control for income. It is become it mediates the effects of educational attainment, and it would be a case of over-adjustment.
· Under “race and ethnicity” it is not clear (as stated) if they will use 2 or 4 groups (this only becomes evident once one arrives at the Results section). It is not clear if there are Hispanic and non-Hispanic blacks as well as Hispanic and non-Hispanic whites.
There are two types of doing race /ethnicity research. Conceptualizing people as groups: non-Hispanic blacks, Hispanic Blacks, Hispanic Whites, and non-Hispanic whites. And studies that deal race and ethnicity as variables rather than groups. Authors have taken both of these approaches in other papers, but because we were interested in the interaction between race/ethnicity and educational attainment, we conceptualized race/ethnicity as variable rather than group.
· It is different to run an analysis on non-Hispanic blacks and non-Hispanic whites than running the analysis on all blacks and all whites controlling for Hispanic origin (which is the approach here). At least as a robustness check, they should run their analysis using non-Hispanic blacks and non-Hispanic whites.
Thanks. Yes. Of course it is different, and we acknowledge this difference. To address this issue, we took two steps. First, we explained in the paper that there the results presented is not limited to Non-Hispanics, and second, we have conducted a robustness check, as the reviewer suggested, and then we added that the robustness analysis using non-Hispanic blacks and non-Hispanic whites does not change the interaction between race and parental educational attainment.
· The authors state their main predictor of interest is SES (line 109). Yet, if the authors have an interactive term, they cannot differentiate the effect of parental education from that of race. This means that their main predictor is not SES alone, but the interaction itself.
With all respect, our main interest is the interaction, but the main independent variable (IV) is not an interaction. So, we kept the language as is. This is how we have been thinking about MDRs, that race (moderator) changes the effect of resource (IV).
· Also, they do not have a variable for “SES”. Their variable is parental education, not SES.
We are no more referring to SES like it is a variable. Now we are more careful about it.
· Does the “ethics” section need to be in the body of the text? It usually goes at the end of the manuscript, and it conforms to stipulations of the publishing journals, not to stipulations of the authors (unless they themselves generated and collected the data).
We moved the ethics to the end part of the paper.
· Another confusion about Table 3 is that it has a column for “B” which I imagine this is the coefficient (?), and also a column for “OR” for an odds ratio… of what? This is not clear.
Thanks. A very relevant and good point. We explained that b is the regression coefficient. OR is deleted.
· Reorganize Table 3: First list the coefficients of interest, then the confounders.
We changed the order of our variables in the tables.
· The interpretation of the interaction in Model 2 is confusing (lines 151-155). In particular this phrase: “This was evident by a negative b for the main effect of parental educational attainment and then a positive b for the interaction term between race and parental educational attainment.”
Thanks. Yes. We have changed this language.
§ The phrase says “[…] and then a positive…”. It seems as if, first a model was fitted and then another (?). Or is it that the authors are going from Model 1 to Model 2 (?). This is confusing.
· Thanks. We are no more referng to first and then. Just to avoid confusion. But yes, we were referring to going from Model 1 to Model 2.
·
· The multivariable models are “by” not “in” race/ethnic groups (line 160).
Thanks. fixed.
§ I don’t see what additional information these models bring aside from that already in Table 3. The difference is that, models in Table 4, are “full-interactive” models whereas those in Table 3 are not. Models in Table 4 may be used as robustness check; I don’t see a valid justification to include them in the paper. If the authors want to describe the separate effects, and to test for their respective statistical significance, for whites and blacks, they can perfectly do that using Model 2 in Table 3. Plus—again—models in tables 3 and 4 are not comparable because models in Table 4 are fully-interactive. To be explicit: By separating the models in Table 4, they are intrinsically specified as if all variables in the right side of the equation are interacted by the respective race they were ran for (whereas in Table 3 only race and parental education are interacted). The models, therefore, follow different econometric specifications; implications from Table 4 do not map into those from models in Table 3.
Thanks a lot! With all due respect, we disagree regarding this matter. It might be a discipline issue. As group differ a lot, even covariates may operate differently across groups. So, we need to run race specific models to see how each covariate and the independent variable operates across groups. So, we run 4 models, not just 2 models, and all of them contribute to our story.
Theoretical context:
· The Background section is too short. I was felt with the impression it went midway. If space is no constraint, I would advise to expand it.
Thanks. We added to the background section.
· Importantly, the authors want to stress that their results bring implications for a wide range of policy arenas, from public to economic and from social to health-specific policy. I really appreciate this context, especially because research has shown again and again that in spite of good-faith policy intentions, we cannot effectively address health disparities and the differential effect of resources on health without talking about structural forces. Yet, I still believe the authors’ approach is still underdeveloped. Social and health policy is written by politicians, and today’s political environment is highly polarized, there is gridlock in Congress, state legislatures behave along party lines, presidents and presidential candidates capitalize on totally different constituencies, lobbies, and corporate interests, and so forth. The paper would greatly benefit from outlining in more detail a structural context to their findings.
Please see our new discussion that brings political perspective to the paper.
· Why we observe the health differences we observe by race and, specifically, the BMI differences the authors report by parental education and race? Yes, we know the reasons go along racism, discrimination, differential treatment, and so forth. But these are the proximal causes. Again, I think the paper would greatly benefit if the authors address fundamental causes, and engrain their findings within the theoretical framework of the historical, legal, and political determinants of the social determinants of health (like parental education) and how they are differently distributed in the population by race.
Thanks. We added some discussion and citations regarding the relevance of historical, legal, and political determinants.
· There is also research they can incorporate on social stratification and biological mechanisms that respond to stress (e.g., allostatic load, weathering, metabolic syndrome) that fit very well with their findings. And, again, I think the paper would benefit from signaling that social stratification by race is historically, politically, and legally patterned.
Thank you! Good Point. We added considerable literature on allostatic load, weathering, as well as well as processes that link race to political power.
· Lines 114-115: Write “We applied PATH’s survey design weights”.
Done. Thank you!
· Line 59: spacing: MDRs literature[8,15,22] ,
Done. Thank you!
· Line 89: typo: 1,0701 instead of 10,701.
Done.
· Line 95: typo or confusing word choice: “as below”
Done.
Line 148: Write: “were estimated using the complete sample” (not “in the overall sample”). The same goes for Table 3’s title: “using the complete sample”.
Reviewer 2 Report
Manuscript ID ijerph-552841
Unequal Effects of Parental Educational Attainment on Body Mass Index of Black and White Youth
This is an interesting, timely, and well-written paper looking at the moderating role of family SES (parental educational attainment) on body mass index in a large sample of US adolescents. Authors used an appropriate theoretical framework (e.g., Minorities’ Diminished Returns theory) to create the hypotheses and read results. I congratulate authors for their work. I have just some minor revisions to suggest:
Lines 77-78: “we decided to the cross-sectional data and limited our variables to the wave 1 of the PATH study”. I think there is something missing. What authors “decided to”?
Line 89: 1,0701 must be changed in “10,701”
I would suggest a little change in the Table 1, so that under the columns “all/whites/blacks” will compare the sample numbers and percentages (deleting the reference to race, as it is confusing), as follows:
All N = 10,701 | Whites N = 8,678 | Blacks N = 2,023 | ||||
n | % | n | % | n | % | |
Ethnicity | ||||||
………. | ||||||
……….. | ||||||
………. | ||||||
Finally, I would suggest authors to include a Figure in which reporting the relationships between variables analyzed, so that readers may immediately understand what authors are testing.
Author Response
This is an interesting, timely, and well-written paper looking at the moderating role of family SES (parental educational attainment) on body mass index in a large sample of US adolescents. Authors used an appropriate theoretical framework (e.g., Minorities’ Diminished Returns theory) to create the hypotheses and read results. I congratulate authors for their work. I have just some minor revisions to suggest:
Thank you. Your positive feedback means a lot to us.
Lines 77-78: “we decided to the cross-sectional data and limited our variables to the wave 1 of the PATH study”. I think there is something missing. What authors “decided to”?
Thanks! The sentence is improved. Sorry for the inconvenience.
Line 89: 1,0701 must be changed in “10,701”
Changed. Sorry or the typo.
I would suggest a little change in the Table 1, so that under the columns “all/whites/blacks” will compare the sample numbers and percentages (deleting the reference to race, as it is confusing), as follows:
All N = 10,701 | Whites N = 8,678 | Blacks N = 2,023 | ||||
n | % | n | % | N | % | |
Ethnicity | ||||||
………. | ||||||
……….. | ||||||
………. | ||||||
Thank you. Sure, we deleted race from Table 1.
Finally, I would suggest authors to include a Figure in which reporting the relationships between variables analyzed, so that readers may immediately understand what authors are testing.
Thank you. We added a figure that compares the association for Blacks and Whites.
Reviewer 3 Report
METHODS
1. (minor) Line 94: Instead of saying "about 14,000", use the exact number.
2. Provide some justification (and perhaps a reference) for dichotomizing age into 12-15 and 16-18.
3. For BMI, please emphasize that the measure is self-reported.
4. It would be helpful to either expand the conceptual model text or perhaps insert a figure (or supplemental material) to illustrate the model used in this analysis.
More information should be provided about the statistical analysis:
5. Were assumptions for linear regression met, especially with respect to the normality of the residuals?
6. If the point is to assess interaction of education and race, then interaction models in addition to the stratified models "across racial groups" should be conducted.
7. It is unclear what is meant by "as a robustness check, they should run their analysis using non-Hispanic blacks and non-Hispanic whites."
8. (minor) In some places, the terms "Whites" and "Blacks" are capitalized, and in others they are lowercase. Please be consistent throughout the manuscript.
RESULTS
9. In the methods section, it states that the two age groups are 12-15 and 16-18. But in the table, the second group appears to be 16-17. (excludes 18?) Please clarify.
10. Regarding comment 6 above, please state clearly in the methods section what models will be used, since interaction terms were used in one set of models.
11. The first Table 3 should be Table 2.
12. P-values should not be 0.000. Please change all instances to "p < 0.001".
13. Figure 1 has unlabeled axes. Please label the axes in the figures and not in the legend.
14. Figure 1 does not need the green background or the number 6 on the x-axis.
15. The dots on Figure 1 should be made substantially smaller and perhaps jittered so that the reader can potentially identify the dot density on the graphs. As of now, many of the dots appear as one line segment.
16. Line 182: Explain what is meant by "inverse correlation".
17. In the Methods section, the authors state that they checked for collinearity of the predictor variables. How was this conducted? VIF? Other method? Was this done for the stratified models, as well?
18. It is recommended that in the Results, the authors should pick two or three sets of results and show the betas and confidence intervals in the text itself, especially for Table 3.
19. Lines 168-169: Was the robustness check actually completed? If so, what were the results? If not, this sentence could be deleted.
Author Response
Thanks for the great comments. Your points have made the paper much improved. All the changes are in yellow. The graph is exception, that is edited but not highlighted. Please let us know what you think.
Here is a list of changes:
(minor) Line 94: Instead of saying "about 14,000", use the exact number.
Corrected.
Provide some justification (and perhaps a reference) for dichotomizing age into 12-15 and 16-18.This is not our decision. The PATH data (public data) has released age variable as a dichotomous variable.
For BMI, please emphasize that the measure is self-reported.Did. Now it is clear that it is based on self-reported height and weight.
It would be helpful to either expand the conceptual model text or perhaps insert a figure (or supplemental material) to illustrate the model used in this analysis.We expanded the text on the conceptual model.
More information should be provided about the statistical analysis.
We added to the description of our models.
Were assumptions for linear regression met, especially with respect to the normality of the residuals?We have tested the assumptions / requirements of the linear regression models. This includes near to normal distribution of the residuals and lack of collinearity between the predictors.
If the point is to assess interaction of education and race, then interaction models in addition to the stratified models "across racial groups" should be conducted.Yes, they are performed. Please see the description of our modeling approach (newly added to the text).
It is unclear what is meant by "as a robustness check, they should run their analysis using non-Hispanic blacks and non-Hispanic whites."We fixed the sentence. Sorry for the confusion, poor writing. It is now clear.
(minor) In some places, the terms "Whites" and "Blacks" are capitalized, and in others they are lowercase. Please be consistent throughout the manuscript.RESULTS
In the methods section, it states that the two age groups are 12-15 and 16-18. But in the table, the second group appears to be 16-17. (excludes 18?) Please clarify.Fixed.
Regarding comment 6 above, please state clearly in the methods section what models will be used, since interaction terms were used in one set of models.The text of our analysis approach is added, and now it is clearer.
The first Table 3 should be Table 2.
Fixed.
P-values should not be 0.000. Please change all instances to "p < 0.001".
Changed
Figure 1 has unlabeled axes. Please label the axes in the figures and not in the legend.
We have mentioned the x and y axes in the figure legend. We thought it would be redundant information.
Figure 1 does not need the green background or the number 6 on the x-axis.There is no green background. It will not show with the green background. This will be resolved. We edited the graph so there is no number 6 on the x-axis anymore.
The dots on Figure 1 should be made substantially smaller and perhaps jittered so that the reader can potentially identify the dot density on the graphs. As of now, many of the dots appear as one line segment.Thanks. Very helpful point. By editing the graph, this is resolved. Now the dots seem more separate.
Line 182: Explain what is meant by "inverse correlation".Negative correlation.
In the Methods section, the authors state that they checked for collinearity of the predictor variables. How was this conducted? VIF? Other method? Was this done for the stratified models, as well?
This is added to the text of the paper.
To rule out multi-collinearity between our study variables, we applied Pearson correlation test between all the study variables. No two variables had correlation coefficients larger than 0.50 so we ruled out any collinearity between our independent variables. This strategy was taken for the full sample and also racial groups.
It is recommended that in the Results, the authors should pick two or three sets of results and show the betas and confidence intervals in the text itself, especially for Table 3.We showed the Bs in the text.
Lines 168-169: Was the robustness check actually completed? If so, what were the results? If not, this sentence could be deleted.Yes, the robustness check was completed. We think they are not necessary because they just confirm / support our main findings. So, we have just added one sentence that it did not change our findings.
Round 2
Reviewer 1 Report
The theoretical component, interpretation of results, and discussion components of the paper improved substantially. I am happy with the revisions. The paper deserves to get published after just a minor revisions:
· There are some spacing issues in the tables that need to be formatted during the final editing process (for example, at the footnotes).
· In Table 1, the tests reported at the bottom are not specific as to what is it that they are testing: Is it the difference in the means between whites and blacks, for example? There are typos, as well, in the footnotes (e.g., “independent” without capitalization).
· In the figures: What is the Y-axis? What the X-axis?
o There is also the issue of zooming in the figures. This is due to displaying the scattered data instead of just the simulation. The scattered data expands throughout the Y-axis, thus it is almost impossible to notice if the lines are increasing/decreasing/stable. That’s because the scale of the rate of change is small compared to the overall scale of the Y-axis. Please get rid of the scattered data, and show one, not two, figures, showing the differences for black and white youth. If you use one instead of two figures, it is also possible to see the interaction between the variables—which is exactly the hypothesis that the authors want to test.
Author Response
Greetings!
Thanks to the great comments by the reviewer.
We are very happy to see that the paper is found much stronger.
This was not possible without the reviewer comments.
We have made modifications based on one of the reviewer comments. He also says :
The paper deserves to get published after just a minor revision:
So, we assume the 1st minor comment is the main one.
We took care of it. Please see the changed to the table descriptions
However, we could not merge the graphs to 1. We tried, but it did not work for us.
So, we think the graphs should stay as is, if this is OK with the editor. This seems not to be very important to the reviewer has he calls it minor revision, and lists as the 2nd one.
Please see the changes in the paper. All the changes are in Yellow.
best
Shervin
